# A Telemedicine Approach to Covid-19 Assessment and Triage

**DOI:** 10.3390/medicina56090461

**Published:** 2020-09-10

**Authors:** Allison B. Reiss, Joshua De Leon, Isaac P. Dapkins, George Shahin, Morgan R. Peltier, Eric R. Goldberg

**Affiliations:** 1Department of Medicine, NYU Long Island School of Medicine, Mineola, NY 11501, USA; Joshua.DeLeon@NYULangone.org; 2Department of Foundations of Medicine, NYU Long Island School of Medicine, Mineola, NY 11501, USA; Morgan.Peltier@NYULangone.org; 3Department of Population Health and Department of Internal Medicine, NYU Langone Health, New York, NY 10016, USA; Isaac.Dapkins@nyulangone.org; 4Department of Internal Medicine, NYU Langone Health, New York, NY 10016, USA; George.Shahin@nyulangone.org; 5Department of Medicine, NYU Grossman School of Medicine, New York, NY 10016, USA; Eric.Goldberg@nyulangone.org

**Keywords:** Covid-19, telehealth, pneumonia, cough, novel coronavirus, droplet

## Abstract

Covid-19 is a new highly contagious RNA viral disease that has caused a global pandemic. Human-to-human transmission occurs primarily through oral and nasal droplets and possibly through the airborne route. The disease may be asymptomatic or the course may be mild with upper respiratory symptoms, moderate with non-life-threatening pneumonia, or severe with pneumonia and acute respiratory distress syndrome. The severe form is associated with significant morbidity and mortality. While patients who are unstable and in acute distress need immediate in-person attention, many patients can be evaluated at home by telemedicine or videoconferencing. The more benign manifestations of Covid-19 may be managed from home to maintain quarantine, thus avoiding spread to other patients and health care workers. This document provides an overview of the clinical presentation of Covid-19, emphasizing telemedicine strategies for assessment and triage of patients. Advantages of the virtual visit during this time of social distancing are highlighted.

## 1. Introduction

Clinical evidence has demonstrated that the Covid-19 virus is easily transmissible from person to person through respiratory droplets from an infected person that are released into the air by sneezing, coughing or talking around other persons [1]. The viral particles can continue to be viable in aerosol form or on surfaces for hours after release [2]. A person’s hands may pick up the virus and, through direct contact with one’s own nasal mucosa, oral cavity or eyes, potentially lead to self-infection [3]. The highly contagious nature of Covid-19 makes the use of telemedicine particularly advantageous because it circumvents risk of spread by contact. Self-presentation to the emergency department is discouraged unless symptoms are severe. Telemedicine or “healing at a distance” consists of remote delivery of care by health professionals through the use of information and communication technologies and can be used for screening, communication, assessment, counseling and monitoring [4,5].

Telemedicine has been defined by the World Health Organization as “delivery of health care services, where distance is a critical factor, by all health care professionals using information and communication technologies for the exchange of valid information for diagnosis, treatment and prevention of disease and injuries, research and evaluation, and for the continuing education of health care providers, all in the interests of advancing the health of individuals and their communities” [6]. It can be accomplished through various platforms in the form of video, audio or text. Telehealth allows large numbers of patients to be assessed while eliminating exposure risk for health care workers [7]. In light of the Covid-19 pandemic, the need for telemedicine to maintain “social distancing” and minimize virus transmission is expected to surge and this creates a need to provide guidance to clinicians who are new to this mode of practice [8,9,10]. If patients are concerned that they have Covid-19, they can be guided in a video call through a symptom check. Determination of a significant risk should be based on presenting symptoms, history, underlying conditions and community transmission of disease. If it is determined that significant risk is present, then a laboratory test may be ordered.

## 2. Covid-19 Presentation

The spectrum of symptoms in persons infected with Covid-19 ranges from asymptomatic to severe and life-threatening [11,12,13]. The infection initially may present with a syndrome similar to a common cold or flu with fatigue, dry cough and sometimes fever and over 80% of patient will have mild, limited disease [14,15]. Acute onset of loss of smell and taste can be presenting symptoms, even before cough and/or fever are noted and may be a useful screening criterion [16,17,18]. Non-specific symptoms that have been reported include sore throat, nasal congestion, rhinorrhea, and myalgia. In approximately 10% of cases gastrointestinal symptoms such as loss of appetite, diarrhea, nausea, or vomiting present 1 or 2 days before fever and cough, and virus can be shed in the stool throughout the disease period [19,20,21]. When presentation is gastrointestinal in nature, diagnosis may be delayed and viral clearance may be slower to occur compared to those with only upper respiratory symptoms [22,23].

Covid-19 is presumed to be infectious 2 to 3 days prior to the onset of symptoms and throughout the course of the illness and infectivity is thought to correspond to detection of virus in nasal, oral and sputum specimens [24,25]. Infected individuals who remain asymptomatic may also be able to transmit the virus [26,27].

## 3. Criteria for Determining Severity

Covid-19 severity levels can be categorized as mild, moderate, severe, and critical conditions (Figure 1) [28,29]. Mild disease is defined as patients without dyspnea, without clinical evidence of respiratory distress, no pneumonia on imaging, and blood oxygen saturation maintained above 93% under resting conditions [30,31]. Fever (<39.1 °C) and cough are seen frequently, even in mild disease. Moderate disease is diagnosed by fever or respiratory symptoms with pneumonia while severe disease manifests as dyspnea and/or hypoxemia. There is respiratory distress with respiratory rate ≥ 30 breaths/min; SpO2 ≤ 93% at rest, and PaO2/FIO2 ≤ 300 [32]. The most critical patients develop acute respiratory distress syndrome (ARDS) with respiratory failure in need of mechanical ventilation, multiple organ dysfunction (MODS) and/or shock, metabolic acidosis, and coagulation abnormalities that are resistant to treatment [33,34].

## 4. Covid-19 Risk Factors

Covid-19 disproportionally affects elderly persons and those over the age of 85 are at highest risk of death [35,36,37] (Figure 2). Infection may spread rapidly among older adults residing in long-term and skilled nursing care facilities and a significant number of Covid-19-related deaths have occurred in residents of these facilities [38,39]. Higher mortality rates have been seen in males in multiple countries [40,41].

Preexisting conditions such as diabetes mellitus, cardiovascular disease, hypertension and obesity also increase the risk of death [42,43,44,45]. Diabetes and cardiovascular disease are found disproportionately in severe cases of Covid-19 requiring ICU admission [46,47,48,49]. Obesity may compromise ventilation at the lung bases and can be considered a state of low grade inflammation and both these factors may contribute to risk of severe Covid-19 infection and greater need for mechanical ventilation in obese patients, especially in patients under age 65 [42,50]. Chronic lung disease and pre-existing chronic obstructive pulmonary disease (COPD) increase the likelihood of developing severe Covid-19 [51,52,53]. A history of cancer is associated with higher incidence of Covid-19 infection along with elevated risk of severe-associated events [54,55,56].

## 5. Laboratory Findings and Imaging

Reverse-transcription polymerase chain reaction (RT-PCR) of oropharyngeal swab samples from the upper respiratory system is the basis for making a definitive diagnosis of Covid-19. This method amplifies specific segments of the Covid-19 genome, thereby detecting the presence or absence of viral nucleic acid. It is used for screening and as the gold standard for diagnostic purposes [57,58,59]. However, false negatives can occur so that a negative result does not exclude infection.

Total white blood cell count in early peripheral blood is normal or decreased. The most common laboratory abnormality in Covid-19 patients is lymphopenia, with decrease in lymphocyte count more profound in severe cases [60,61]. Inflammatory markers such as C-reactive protein, ferritin and IL-6 are increased in most patients [62].

Serum sedimentation rates and high sensitivity C-reactive protein are often increased and greater elevations in these markers of inflammation may be associated with disease severity [63,64,65,66]. Biochemical test results may show an elevated plasma D-dimer level and D-dimer above 1 μg/L is indicative of hypercoagulability and a poor prognostic indicator [67,68,69,70,71,72].

Covid-19 pneumonia typically presents on CT scans as a bilateral ground glass appearance, with or without consolidation [73,74,75]. In Covid-19, lesions are often distributed in peripheral and subpleural areas of the lung [76]. However, the findings are non-specific and may be present in other types of viral pneumonias. The negative predictive value of CT in a multicenter study in China was only 42%, and since in a substantial portion of COVID-19 patients with minimal symptoms a chest CT may be normal, CT is not recommended as a screening tool for Covid-19 [77,78].

## 6. The Telemedicine Evaluation

This section describes, in general terms, the telemedicine evaluation of a potential Covid-19 patient as practiced in the offices of physicians affiliated with a major urban medical center.

### 6.1. Initial Screen

If a patient presents in-person at the office, screening will take place to determine whether the patient can enter the facility. The patient is screened at the door by a screener in full PPE. If the patient screens positive—schedule a virtual visit (same day).

The screener will follow the protocol outlined:
*(1)* Temperature Check

Temperature checked at time of arrival for all patients and approved visitors: If 100.0 °F or greater, patient/visitor is not permitted to enter the clinic and will be instructed to return home (unless in obvious distress) and provided information to schedule a virtual visit. If less than 100.0 °F, proceed to screening questions.

*(2)* 
*Covid-19 Screening Questions*


Our practice utilizes a set of screening questions (Table 1). Patients are informed that their responses will be kept confidential. Clarification may be needed in cases where patients report symptoms that are not of recent onset and are associated with chronic health conditions.

### 6.2. Virtual Visit

#### 6.2.1. Introduction, Consent, Symptom Check, Demographics

Introduction: The physician greets the patient and introduces themselves. The patient’s location, name and date of birth are confirmed. The patient is asked whether an interpreter is needed and, if so, an interpreter is provided via a certified interpretation service (conferenced into the call).Consent: It is explained to the patient that to provide necessary care the virtual visit will be conducted as a replacement for an onsite visit in order to maintain the patient’s safety and the safety of our staff. Verbal consent to proceed with the virtual visit must be obtained. The provider advises the patients of the risks and benefits of the virtual visit. If the patient consents, then the provider documents that the patient understands the risks and benefits of the virtual/telephone visit as discussed and consents to the visit.Vital Information—It is helpful if the patient is able to assess their temperature and oxygen saturation. To assess oxygen saturation patients discharged with home oxygen are provided a pulse oximeter at the time of discharge.Symptoms—The assessment of the patient’s symptoms are based on Centers for Disease Control (CDC) guidelines [79]. This assessment is similar to the assessment done at the time of Covid-19 screening. Key symptoms that raise the index of suspicion for Covid-19 infection is answering “Yes” to ANY ONE of the highest priority questions or ANY TWO of the high priority symptoms in Table 1.Assessing clinical stability—it is important to identify patients who need an immediate onsite evaluation at a designated screening center or a hospital emergency department. These are locations where in-person Covid-19 evaluation and testing can take place. The following questions help determine which patients are unstable and need an immediate in-person evaluation:
Is the oxygen saturation less than 90%?Is the temperature greater than or equal to 102 °F and not responding to antipyretics?

If symptoms suggest respiratory compromise or hypoxia and the patient is determined to be unstable, then the healthcare provider will instruct the patient to call 911 to go immediately to the emergency department. If the patient is not unstable, but the healthcare provider has determined that an onsite in-person evaluation is warranted, then the patient is referred to a designated screening center.

Demographic data—data collected is based on factors that have been shown to be related to a higher incidence and/or severity of Covid-19 disease and must be taken into account in risk assessment as noted below.
○Age—greater than 65 are considered “vulnerable”○Sex—males affected greater than females.○Country of origin or race if relevant—endemic areas with high incidence. African-Americans and Hispanics are disproportionately affected.○Travel history—pertinent if traveled to an endemic area. Recent travel (within the last 14 days)—Either international travel or travel within the United States, dates of travel, and destination.

#### 6.2.2. The Telemedicine Medical History

Chief complaint: reason for the visit in the patient’s own words.History of present illness (HPI): What symptoms are you experiencing and how long have you had these symptoms?In addition to the symptoms covered earlier, are you experiencing any of the following?
○Diarrhea○Abdominal pain○Nausea or vomitingSignificant past medical history (PMH) (Table 2)List current medicationsAllergiesFamily history

#### 6.2.3. The Telemedicine Physical Exam

The physical exam conducted virtually is patient/caregiver facilitated via video observation (Table 3). The patient or another member of household may take as many vital signs as possible, including temperature, body weight, blood pressure, heart rate.

## 7. Laboratory Testing

Not all patients will require blood tests. However, in the event that a patient requires lab testing it should be available, by appointment. In the follow-up plan, the provider should note that the patient needs to be scheduled for labs. The support staff working with the provider can then contact the patient to schedule an evaluation.

Lab tests relevant for these patients include: complete blood count, comprehensive metabolic panel, D-dimer [67], erythrocyte sedimentation rate [64], C-reactive protein [62], and Covid-19 IgG. Creatine kinase-MB fraction and troponin may also be measured as markers of possible myocardial injury [80,81].

## 8. Patients Requiring Self-isolation or Quarantine

Patients who meet criteria for a diagnosis of Covid-19 infection not requiring emergency medical attention are instructed to go home and self-isolate or self-quarantine.

According to the CDC, isolation separates sick people from people who are not sick while quarantine separates and/or restricts the movement of people who were exposed to a disease. Both are ways to prevent the spread of an infectious disease like Covid-19 [82].

Below are the instructions provided to our patients to protect themselves and others.

### 8.1. How to Self-Isolate/Quarantine

Stay at home. Take every possible step to reduce going into public spaces. Avoid contact with others. Do not let anyone visit you in your home until your self-isolation/quarantine is over.Practice social distancing. If you have to leave your home, practice social distancing. This means trying to maintain a six-foot distance from others. Avoid using any kind of public transportation whenever you can. This includes ridesharing services and taxis.Practice good hygiene. Always cover your nose and mouth when you cough or sneeze. Use the crook of your arm (inside your elbow). Do not cough and sneeze into your hand. Wash your hands afterwards.Wash your hands. You should be washing your hands often with soap and water for at least 20 s. If soap and water are not available, use an alcohol-based sanitizer. Make sure that it contains at least 60% alcohol.Avoid sharing personal items. Do not share eating utensils and other personal items such as toothbrushes, drinking glasses, and water bottles.Clean surfaces. Disinfect surfaces you touch often such as cell phones, doorknobs, light switches, counters, tabletops, etc. Wash your clothes, bath and kitchen towels, clothes you sleep in, bedsheets and pillowcases regularly.Rest and drink plenty of fluids to stay hydrated.Monitor your health. If you develop any symptoms or they get worse, call your health care provider or make an appointment with your virtual urgent care provider.

### 8.2. If You are Having Any Symptoms Such as Fever or Cough Follow These Additional Instructions:

If there is an older adult (over the age of 65) in your home with organ failure, a weakened immune system or uncontrolled diabetes, this person should not share living space with you. If that is not possible, avoid close contact. Have separate sleeping arrangements. Prepare and eat meals separately as well. Use a separate bathroom if possible.If you must share a living space with high-risk family members, wear a mask when you are near them. Have as little contact with these people as possible until you no longer have a fever and cough.If you must leave your home and you are having any symptoms, wear a face mask, goggles and/or face shield.

### 8.3. When Can You Stop Self-Isolation/Quarantine?

According to CDC guidelines, you can stop self-isolation/quarantine if all of the following applies to you:
You have had no fever for 72 h (three full days) without the use of fever reducing medicines
and
Other symptoms such as cough or shortness of breath have gotten better or are almost gone
and
At least seven (7) days have passed since you first started having symptoms. Call your healthcare provider with any questions or if you are unsure whether you can stop self-isolation/quarantine

## 9. Management

Mildly symptomatic Covid-19 is a self-limiting illness and management consists of supportive care with rest, fluids, and antipyretics combined with close monitoring for clinical deterioration [83]. There is no safe, effective and proven treatment at this time. An optimal, evidence-based approach to averting a severe inflammatory response is needed and efforts are ongoing to develop a strategy to achieve this [84]. Of course, vaccine development is the highest priority [85].

## 10. Limitations of Telemedicine

The advantages of telemedicine in assessing and managing Covid-19 have been highlighted here, but when deciding whether this approach is the right one for an individual patient, it is important to consider the drawbacks. Telehealth is only possible if the patient has literacy in the modality used for delivery and if the internet or phone connection is of reasonable quality. Bandwidth, software or other technical issues may interfere with data transmission and obstruct visual and/or auditory aspects of communication [86]. This problem may be encountered more commonly in rural areas and in socioeconomic disadvantaged environments with limited access to technology [87]. Privacy and confidentiality may also be an issue for patients using equipment in areas frequented by other household members. Use of headphones by the patient may be helpful, but do not guarantee privacy. Persons with barriers to use of technology such as visual or hearing impairment may require in-person visits, although specialized communication platforms can make telecare feasible in some circumstances [88,89]. Without the in-person encounter, the feeling of a personal connection and establishment of a provider-patient relationship with the key elements of trust and mutual respect is more difficult [90,91,92].

## 11. Conclusions

Our offices in the New York City area are using live interactive telemedicine during the Covid-19 pandemic to provide medical screening and assessment remotely. The telehealth platform allows for delivery of care while maintaining the physical distancing necessary to prevent the spread of this infectious disease [93]. Ultimately, widespread adoption of this technological tool will be determined by formal studies of quality, comparing telehealth versus in-person outcomes. This type of analysis has just begun and early results are promising with indications that telehealth is reaching persons who might have sought no care at all without this option [94,95].

## Figures and Tables

**Figure 1 medicina-56-00461-f001:**
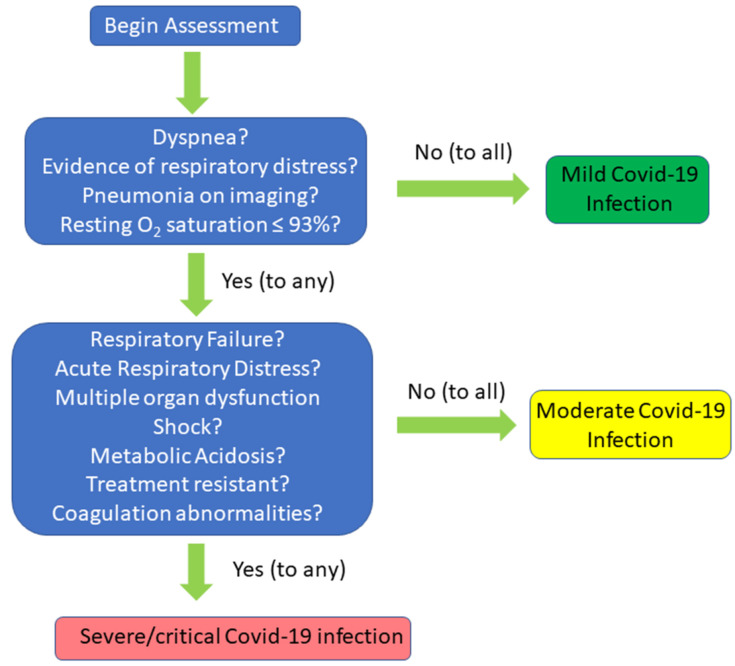
Decision-tree diagram for the classification of severity of Covid-19 infection.

**Figure 2 medicina-56-00461-f002:**
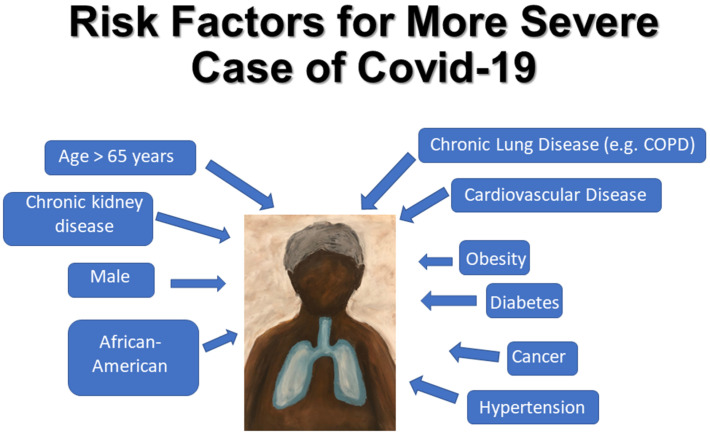
Diagram of risk factors for developing severe Covid-19 infection. COPD, chronic obstructive pulmonary disease.

**Table 1 medicina-56-00461-t001:** Covid-19 Screening Questions.

**If “Yes” to ANY ONE of the following highest priority questions, the patient is not permitted to enter the clinic.**
(1) Have you had a temperature of 100 °F or greater in the past 7 days?
(2) Have you been diagnosed with Covid-19 in the past 14 days?
(3) Have you had contact with a known confirmed Covid-19 positive person in the last 14 days?
(4) Do you have a cough?
(5) Do you have shortness of breath or difficulty breathing?
**If “Yes” to having ANY TWO of the following high priority symptoms in the past 14 days, then the patient is not permitted to enter the clinic. The patient/visitor will be provided with the information to schedule a virtual visit:**
(1) Fever
(2) Cough
(3) Repeated shaking with chills
(4) Muscle pain
(5) Headache
(6) Sore throat
(7) New loss of taste or smell

**Table 2 medicina-56-00461-t002:** Covid-19 Past Medical History.

Do you have any of the following pre-existing medical conditions?
Chronic lung disease
COPD
Asthma
Diabetes (type I or II)
History of heart disease, myocardial infarction, angioplasty, stent placement, coronary artery bypass graft
Valvular heart disease
Chronic kidney disease Chronic liver disease Immune system disorder or compromise Past surgeries

COPD, chronic obstructive pulmonary disease.

**Table 3 medicina-56-00461-t003:** The Telemedicine Physical Exam.

General. Look for: any apparent distress, toxic/nontoxic appearance
Head. Normocephalic. atraumatic
Eyes. Observe whether extraocular movements are intact; look for scleral icterus
Ears. Appearance of external ears
Throat. Facilitated: oropharyngeal viewing with flashlight
Neck. Observe for full range of motion and midline trachea
Respiratory. Is there distress or use of accessory respiratory muscles?
Abdomen. Facilitated: nondistended and nontender?
Neurologic. Check for alertness without focal deficits, observe gait
Psychologic. Mood and affect; assess for appropriate relation with provider via camera

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
