# Peer review of "A Telemedicine Approach to Covid-19 Assessment and Triage"

_medicina, 2020, doi:10.3390/medicina56090461_

Round 1

Reviewer 1 Report

 The authors have reviewed the use of telemedicine for patients with COVID-19.

Comments

  • This article focuses on the topic of telemedicine for patients with COVID-19. They provided practical information such as a decision-making tree and screening checklist.
  • I understand that telemedicine is very convenient way for the patients with COVID-19. However, I recommend arguing the weakness or limitation of telemedicine compared to the physically visiting.
  • The authors should consider and argue the patients with difficulty in continuing telemedicine. They have several reasons; 1) Some patients are unable to use a video-call system for a variety of reasons, such as difficulty accessing the internet, location, or financial situation., 2) People who are blind or hard of hearing may be more inconvenienced by telemedicine.
  • Both figures which authors provided should be of higher resolution, especially Figure 1.

Author Response

Reviewer # 1 Comments

1) COMMENT #1:   I understand that telemedicine is very convenient way for the patients with COVID-19. However, I recommend arguing the weakness or limitation of telemedicine compared to the physically visiting.

The authors should consider and argue the patients with difficulty in continuing telemedicine. They have several reasons; 1) Some patients are unable to use a video-call system for a variety of reasons, such as difficulty accessing the internet, location, or financial situation, 2) People who are blind or hard of hearing may be more inconvenienced by telemedicine.

RESPONSE: We appreciate this very constructive criticism and completely agree that to add balance to the paper we need to discuss the shortcomings of telemedicine. We have therefore added an entire section describing the difficulties entitled “Limitations of Telemedicine”

2) COMMENT #2:   Both figures which authors provided should be of higher resolution, especially Figure 1.

RESPONSE: We have fixed the issue and submitted higher resolution versions of the figures.

Reviewer 2 Report

The authors describe the background of COVID-19 and provide a process level description of a telehealth approach for screening patients. The telehealth approach is safer than screening in-person. 

Overall, the paper is well-written and the use of telehealth for COVID-19 screening is important. 

However, I do not find the article and the process described particularly novel. Most healthcare systems in the United States and internationally are using a telehealth approach and there have been several publications on the use of telehealth for COVID-19 screening.  

Mann, D. M., Chen, J., Chunara, R., Testa, P. A., & Nov, O. (2020). COVID-19 transforms health care through telemedicine: evidence from the field. Journal of the American Medical Informatics Association.

Ratwani, R. M., Brennan, D., Sheahan, W., Fong, A., Adams, K., Gordon, A., ... & Booker, E. (2020). A descriptive analysis of an on-demand telehealth approach for remote COVID-19 patient screening. Journal of Telemedicine and Telecare, 1357633X20943339.

Author Response

Reviewer # 2 Comments

1) COMMENT #1:   However, I do not find the article and the process described particularly novel. Most healthcare systems in the United States and internationally are using a telehealth approach and there have been several publications on the use of telehealth for COVID-19 screening.

Mann, D. M., Chen, J., Chunara, R., Testa, P. A., & Nov, O. (2020). COVID-19 transforms health care through telemedicine: evidence from the field. Journal of the American Medical Informatics Association.

Ratwani, R. M., Brennan, D., Sheahan, W., Fong, A., Adams, K., Gordon, A., ... & Booker, E. (2020). A descriptive analysis of an on-demand telehealth approach for remote COVID-19 patient screening. Journal of Telemedicine and Telecare, 1357633X20943339.

RESPONSE: Both of the papers cited by the reviewer are a description of statistics on use of telemedicine and we have now cited them in our manuscript (references 94 and 95) in an expanded conclusion where we describe the need for formal analysis of the quality of telehealth versus in-person care. We note that neither of these papers guides the healthcare professional in how to actually proceed with the visit itself. Our paper is unique because it serves as a practical source of basic information on conducting the actual visit step-by-step. We think that many healthcare providers worldwide will be interested in seeing this in a compact, written format as they undertake a methodology that is new for them.  Even those who are not using telemedicine directly will gain insight into the experience that can help them to decide whether it is a viable option for their future use or for referral of their patients under compelling circumstances. We have now further distinguished this paper by adding a new section describing the limitations of the telemedicine approach.

Round 2

Reviewer 1 Report

Thank you for reflecting my point.

Reviewer 2 Report

The authors have made important revisions to the manuscript. I still question the contribution of this paper to the literature. Perhaps this content is better disseminated as a "how to guide" or a toolkit, not a journal publication?